# Food handlers' knowledge, attitudes and self-reported practices regarding safe food handling in charitable food assistance programmes in the eThekwini District, South Africa: cross-sectional study

Sizwe Earl Makhunga ,[1] Margaret Macherera,[2] Khumbulani Hlongwana[1,3]

¹Public Health, University of KwaZulu-Natal College of Health Sciences, Durban, KwaZulu-Natal, South Africa
²Department of Crop and Soil Sciences, Lupane State University, Bulawayo, Zimbabwe
³Cancer and Infectious Diseases Epidemiology Research Unit (CIDERU), University of KwaZulu-Natal, Durban, South Africa

**Correspondence to**
Sizwe Earl Makhunga;
951030918@stu.ukzn.ac.za

## ABSTRACT

**Objective** This study aimed to assess food handlers' knowledge, attitudes and self-reported practices towards safer donated foods.

**Design** This cross-sectional study was conducted from January to May 2021.

**Setting** This study was conducted in the eThekwini District, South Africa.

**Participants** A convenience sampling-based cross-sectional survey of food handlers (n=252) served as the study's main source of data. A total of eight study sites across five municipal planning regions of eThekwini District were visited. The principal investigator administered the validated structured standardised questionnaire, using participants' language of choice, which was either English or isiZulu. All the questions in the questionnaire were asked in exactly the same manner, following the same flow, in order to avoid bias and ensure consistency. Furthermore, the close-ended nature of questions in the questionnaire mitigated the risks of question manipulation. The questionnaire was adapted from the WHO's five keys to a safer food manual.

**Primary and secondary outcome measures** The primary outcome was the knowledge, attitudes and self-reported practices towards safer donated foods. The secondary outcomes were the sanitary conditions of infrastructure and facilities, food handlers' observed food handling behaviour and nutritional information/labelling.

**Results** The results show that the food handlers had good knowledge, positive attitude and acceptable behaviours towards safe food handling and general hygiene, with mean scores (SD) of 62.8 (14.6), 92.5 (14.1) and 80.4 (13.3), respectively. Significant correlations were found between knowledge and attitudes (p=0001), knowledge and practices (p<0001), and attitudes and practices (p=0.02). However, the correlations between knowledge versus attitude and attitude versus practice were poor (Spearman's r<0.3), and the association between knowledge versus practice was moderate (0.3–0.7). The majority of food handlers (92.5%) did not understand the value of thorough cooking and temperature control. About 53% of respondents acknowledged to never adequately reheating cooked meals, 5% did not see the significance of preventing cross-contamination and 5% were unsure.

**Conclusions** Despite the relatively positive knowledge, attitude and practice levels of the food handlers, safe food handling and hygiene practices, such as thorough cooking and temperature control, properly reheating cooked meals and taking precautions to prevent cross-contamination, require some emphasis.

---

### STRENGTHS AND LIMITATIONS OF THIS STUDY

⇒ The sample size of the study was not too large. However, the diverse charitable food assistance programmes (CFAPs) from which the sample was drawn offer unique strength to the study.

⇒ The study was prone to social desirability bias, given that most data emanated from self-reporting.

⇒ Even though the sampling was thorough and included every CFAP in the database, it is possible that some participants were left out.

⇒ The walkthrough survey, which was used to collect observational data, in addition to survey data in order to supplement survey results and lessen the impacts of social desirability bias, was not included in the study.

## INTRODUCTION

Foodborne diseases (FBDs) resulting from contaminated food continue to pose significant public health challenges worldwide, particularly in low/middle-income countries (LMICs), where socioeconomic inequities are pervasive.[1] FBDs are a diverse group of illnesses caused by eating foods contaminated with infectious or poisonous microbes and/or chemicals.[2] Pathogenic microorganisms, such as bacteria, viruses, parasites, fungi and chemicals, are the primary causes of FBDs.[3] In the clinical environment, FBDs are most exhibited as gastrointestinal symptoms.

Other symptoms, such as neurological, immunological and gynaecological issues, may also occur.[2]

While unsafe food is a global health hazard, infants, small children, the elderly, pregnant women and persons with underlying illnesses are most susceptible, putting them at greater risk of severe FBD.[4–6] FBDs are disproportionately distributed globally, with Africa having the highest rate of FBD per capita globally. Infections associated with diarrhoeal diseases account for a significant proportion of the cases.[4] The high incidence of FBD is due to the usage of unfit water for cleaning and processing food, inefficient food production, poor handling of food, insufficient food storage infrastructure, and regulatory policies that are either ineffective or ineffectively enforced.[1 7 8] The listeriosis outbreak in 2017–2018 in South Africa was a reminder of the potentially catastrophic consequences of the FBD outbreak, underscoring the need to improve food safety controls and interventions.[1 3] Nine hundred and thirty-seven cases were found, 465 (50%) of which were linked to pregnancy, and 406 (87%) of the pregnancy-related cases occurred in neonates. Two hundred and twenty-nine (24%) of the 937 cases involved individuals aged 15–49 years, excluding those who were pregnant. Among the patients whose HIV status was known, 38% of those with pregnancy-related cases (77 of 204) and 46% of the remaining patients (97 of 211) were infected with the *Listeria* virus. One hundred and ninety-three (27%) of 728 patients with a known outcome died.[9–11]

In KwaZulu-Natal (KZN), the health risks associated with unsafe food handling are prevalent, owing to the high incidence of tuberculosis, HIV and AIDS in the province.[12 13] Research has shown that a seemingly minor foodborne infection can be lethal in immunocompromised people.[14] According to the National Institute for Communicable Diseases Review Report on FBD outbreaks, the KZN province had the most (5 cases per 100 000 population) FBD outbreaks between March 2018 and March 2020, outnumbering all nine provinces in South Africa.[7] Research has shown that malnutrition and diarrhoea caused by consuming contaminated food are the leading causes of infant mortality in hygiene-challenged settings like the eThekwini District.[15]

Fifty-four per cent (54%) of the unemployed in the KZN province live in the eThekwini District, thereby making the district vulnerable to health conditions exacerbated by poverty. For example, 31% of the population in the eThekwini District is poor, putting the district at increased risk of food insecurity.[13] As a result, there is a strong reliance on government services and charitable food assistance programmes (CFAPs). For the purpose of this study, CFAPs are described as food establishments that gather and distribute surplus food to people who are food insecure.[9 10] These organisations include food banks, food rescue organisations, pantries, community soup kitchens, child and youth care centres, and emergency shelters.[9 11] Despite several CFAPs operating in the eThekwini District for decades, there is a lack of data about the status of their safety and hygiene practices. However,

a scoping review of studies from high-income countries (HICs) by Makhunga *et al*[16] revealed that the majority of food handlers employed by CFAPs are untrained volunteers who work on a volunteer basis.

Food handlers' knowledge, attitudes and practices towards safe food handling are essential in improving food safety and reducing inadvertent foodborne illnesses.[5 17] Food contamination caused by poor personal hygiene, particularly inefficient handwashing, has been identified as an essential risk factor for food poisoning.[18] Unlike in the conventional food supply chain, there is a lack of data on hygiene practices regarding food handling in CFAPs.[19 20] Thus, this study investigated food handlers' knowledge, attitudes and self-reported practices of food handling on five keys to safer food, based on the WHO's five keys to a safer food manual[21 22] in the CFAPs in eThekwini District, South Africa. The findings of this research can be used to help develop strategies for enhancing food safety in CFAPs and, in turn, protect vulnerable populations from foodborne infections.

## MATERIALS AND METHODS
### The study area
This study was conducted in eThekwini District, South Africa, between January and May 2021. eThekwini District is 1 of the 11 districts in KZN, South Africa. It comprises 103 urban, rural and perirural wards, which spread across eight subdistricts: South Central, South-West, Umlazi and Engonyameni, Lower South, North Central, Greater Inanda/Tongaat subdistrict, Inner West and Outer West. It covers approximately 2556 km$^2$ area stretching from the Umkomaas in the south, Tongaat in the north and Cato Ridge in the west, including a few tribal areas in Umbumbulu and Ndwedwe. It has an estimated population of 3 702 231 (2016 estimate) and a population density of 1448.5 people per km$^2$, with a sizeable population settling in the South Region (41%), followed by the North Region (32%) and the West Region (27%).[13] Despite being highly urbanised and densely populated, pockets of rural communities exist on the west, south and north outskirts, impacting equitable access to essential services.[13]

### Research design and sampling
A convenience sampling-based cross-sectional survey of food handlers (n=252) served as the study's main source of data. A total of eight study settings (South Central, South-West, Umlazi and Engonyameni, Lower South, North Central, Greater Inanda/Tongaat subdistrict, Inner West and Outer West) across five (North, Central, South, Inner West, and Outer West) municipal planning regions of eThekwini District were visited and participated in the study.

### Multistage sampling
#### Stage one: identification of CFAPs
As a first step in creating a database of CFAPs, we checked the website of the National Department of Social Development. We also approached the FoodForward SA Durban

to obtain the list of CFAPs in their database because the process was not exhaustive. The two databases were combined to create a single comprehensive database that had 196 CFAPs. The study included all 196 CFAPs, making it a population sampling.

### Stage two: identification of participants

It was decided that a sample size calculation of 192 was sufficient. We could only get 196 CFAPs, which meant that one participant per CFAP would yield the desired outcome. However, in CFAPs with a high staff volume, we allowed for a maximum of three participants. The convenience sampling approach meant that everyone who met the criteria, was accessible, willing to participate and available was included.

### Data collection

A researcher administered the structured standard questionnaire face-to-face to all sampled and consenting food handlers (n=252) in the selected CFAPs (n=196). A minimum of one and a maximum of three food handlers were recruited in each of the 196 participating CFAPs. The CFAPs that were chosen were visited during their usual business hours. Before participating in the study, each subject signed an informed consent form. The inclusion criteria were all persons involved in food handling. The principal investigator administered the validated structured standardised questionnaire, using participants' language of choice, which was either English or isiZulu.

All the questions in the questionnaire were asked in exactly the same manner, following the same flow, in order to avoid bias and ensure consistency. Furthermore, the close-ended nature of questions in the questionnaire mitigated the risks of question manipulation. The questionnaire was adapted from the WHO's five keys to a safer food manual.[5 23 24] The questionnaire was pilot tested in 10 CFAPs at the Umsunduzi Municipality in Pietermaritzburg, South Africa. Cronbach's alpha coefficient (α) of internal consistency[25] was used to estimate the reliability of the questionnaire, which was found to be acceptable at 0.76. Therefore, since there were no issues with the questionnaire, no changes were required.

### Research instrument

The questionnaire was adapted from the WHO's 'five keys to safer food' guideline for food handlers.[22] We used the manual of the five keys available on the WHO site, first released in a poster in 2001. This handbook includes two sections: section one provides background information for the manual, and section two stipulates five keys to safer food. The participants' food safety knowledge, attitude and self-reported behaviour were assessed using three forms found at the end of section two of the handbook. The questionnaire was structured into six distinctive parts to collect information (1) demographic characteristics, (2) background characteristics of the CFAPs, (3) food hygiene and safety practices and general requirements, (4) knowledge, (5) attitude and (6) self-reported practices. However, this paper focuses on four parts, excluding parts two and three. We also used a checklist to gather data during walkthrough surveys and observations of practices. However, the findings of the walkthrough survey did not form part of this study.

### Statistical analysis

All statistical analyses were performed using Stata Statistical Software V.17 (StataCorp, College Station, Texas, USA). $X^2$ test was used to determine the relationship between the sociodemographic characteristics of the food handlers and their knowledge, attitudes and self-reported practice level. We used descriptive statistics to summarise the CFAPs and food service staff's general characteristics, and describe their food hygiene and safety knowledge, attitudes and self-reported practices. By categorising answers for each section as dichotomous variables, statistical analysis of the relationship between questionnaire answers and demographic characteristics of respondents was performed: knowledge was classified and recorded as correct versus incorrect, attitudes as agreement versus disagreement/uncertain, and practices as safe, when an answer was always, versus unsafe, when an answer was often or never.

Only the questions with a proportion of accurate answers of 95% or less were reported. The level of knowledge of the food handlers was determined through the following criteria: each correct response received 1 point, while each incorrect answer received 0 points. The score allocation for attitude was 2 points for agreeing, 1 point for not sure and a 0 for disagreeing. Based on the response to 11 knowledge questions, the scores could vary from 0 to 11. Three levels were considered for this: scores between 0 and 4 were classified as poor, 5 and 7 as moderate, and 8 and 11 as good. Total scores could differ for 10 attitude questions, and 0–20 classifications were allocated: 0–6 being negative, 7–13 being neutral and 14–20 being positive attitude levels. For 10 practice questions, participants could choose from options 1 to 5, where 1 referred to never and 5 to always. Scores were allocated as follows: always (4), mostly (3), sometimes (2), rarely (1) and never (0). Total scores could differ for 10 practice questions, and 0–20 classifications were allocated: 0–6 being poor, 7–13 being moderate and 14–20 being acceptable practice levels.

### Patient and public involvement

Patients and/or the public were not involved in the design, conduct, reporting or dissemination plans of this study.

## RESULTS

### Distribution of study participants among subdistricts

A total of 252 food handlers participated, most coming from the Greater Inanda/Tongaat subdistrict and Lower South (table 1).

**Table 1** Distribution of participating CFAPs (n=196) and study participants (n=252)

| Location | CFAPs (n=196) | Food handlers (n=252) |
|---|---|---|
| South Central | 28 | 40 |
| South-West | 12 | 18 |
| Umlazi and Engonyameni | 14 | 20 |
| Lower South | 42 | 50 |
| North Central | 36 | 40 |
| Greater Inanda/Tongaat subdistrict | 54 | 70 |
| Inner West | 4 | 8 |
| Outer West | 6 | 6 |

CFAPs, charitable food assistance programmes.

### Sociodemographic characteristics of the food handlers

Ninety-four per cent (n=237 of 252) of the food handlers were women, with nearly half (49.2%) being over 50 years (table 2). Less than half (40.5%) of food handlers had a grade 12 (Matric) certification, while most (59.1%) had a post-Matric qualification. The majority (72.2%) of participants had more than 10 years of service experience, and 60.7% related to the food service industry. All food handlers (100%) had direct contact with food during receiving, checking and storage. As high as 80% of them were involved in food preparation and processing/cooking. The majority (88.1%) of participants reported their primary role as serving, packaging and distributing, with only 14.7% washing and cleaning the food facility. Only 12.3% of food handlers had attended at least one official training session on food hygiene elements within 5 years preceding this study, while the majority (87.7%) reported to have had no training in food safety (table 2).

### Knowledge

All food handlers (100%) knew it was essential to wash their hands before handling food (table 3). About 85% of food handlers were aware of the dangers of food cross-contamination and consequently reported that they had taken steps to avoid it. However, 17% of food handlers were unaware that wiping cloths can spread bacteria, primarily when used filthy. Most food handlers (71%) were enthusiastic about safe food handling. However, 53% admitted never properly reheating cooked meals when asked about their food handling methods. Most (92.5%) of the food handlers did not appreciate the importance of thorough cooking and temperature control. Although almost all food handlers (99.6%) reported that they wash fruits and vegetables, the majority (90.7%) thought one could tell water safety by simply looking at it. The overall results demonstrated that about 41% of food handlers in CFAPs have poor knowledge of safe food handling and general hygiene.

**Table 2** Demographic characteristics of food handlers (n=252)

| Sociodemographic characteristics | Frequency (n) | Percentage |
|---|---|---|
| **Gender** | | |
| Male | 15 | 6.0 |
| Female | 237 | 94.0 |
| **Age group (years)** | | |
| 20–29 | 14 | 5.5 |
| 30–39 | 36 | 14.3 |
| 40–49 | 78 | 31.0 |
| 50+ | 124 | 49.2 |
| **Race** | | |
| Black | 144 | 57.1 |
| White | 19 | 7.5 |
| Indian | 69 | 27.4 |
| Coloured | 20 | 7.9 |
| **Marital status** | | |
| Single | 106 | 42.1 |
| Married | 107 | 42.5 |
| Divorced | 29 | 11.5 |
| Widowed | 10 | 4.0 |
| **Number of dependents** | | |
| None | 24 | 9.5 |
| <2 | 67 | 26.6 |
| 2–5 | 124 | 49.2 |
| >5 | 37 | 14.7 |
| **Place of residence** | | |
| Formal settlement | 181 | 71.8 |
| Informal settlement | 3 | 1.2 |
| Shelter | 68 | 27.0 |
| **Education level** | | |
| Primary | 1 | 0.4 |
| Secondary | 102 | 40.5 |
| Tertiary | 149 | 59.1 |
| **Overall work experience (years)** | | |
| 3–5 | 18 | 7.1 |
| 6–10 | 52 | 20.6 |
| >10 | 182 | 72.2 |
| **Experience in food service (years)** | | |
| <1 | 6 | 2.4 |
| 1–3 | 23 | 9.1 |
| 4–6 | 53 | 21.0 |
| 7–10 | 17 | 6.7 |
| >10 | 153 | 60.7 |
| **Main job responsibility** | | |
| Food receiving, checking and storage | 252 | 100.0 |

Continued

**Table 2** Continued

| Sociodemographic characteristics | Frequency (n) | Percentage |
|---|---|---|
| Food preparation, process/cooking | 202 | 80.2 |
| Food serving/ packaging/ distribution | 222 | 88.1 |
| Washing and cleaning | 37 | 14.7 |
| Other | 0 | 0.0 |
| Attended safe food handling training | | |
| Yes | 31 | 12.3 |
| No | 221 | 87.7 |
| If yes, the date (years) | | |
| ≤2 | 17 | 6.7 |
| 3–4 | 14 | 5.6 |
| Not applicable (N/A) | 221 | 87.7 |

## Attitude

All the food handlers (100%) agreed that it is crucial to observe general hygiene, and 89.3% believed using different knives and cutting boards for raw and cooked foods is worth the extra effort. However, 5.2% did not show appreciation of the importance of guarding against cross-contamination, and 5.5% were uncertain. Almost all food handlers (96.8%) reported ensuring thorough cooking and observed temperature controls. They were also aware of the consequences of improperly thawing food. Furthermore, 75% of food handlers believe that leaving prepared food at room temperature for more than 2 hours without refrigeration is dangerous (table 4). With a score of 96.8%, the food handlers in CFAPs demonstrated a positive attitude toward safe food handling and general hygiene.

## Self-reported practices

All food handlers (100%) stated that they always maintain a high level of general hygiene (table 5). Nearly two-thirds (65.9%) of the participants used separate surfaces for cooked and ready-to-eat food. However, 84.9% of the participants reported segregating raw and cooked food during storage. Food handlers reported observed temperature control in 68.3% and 55.6% of cases, respectively. Less than half (43.7%) of the participants said it was necessary to reheat cooked food until it was boiling. All food handlers (100%) said they inspect and discard food that has passed its expiration date. They also asserted that they wash fruits and vegetables with clean water before consuming them. The food handlers in CFAPs reported acceptable general hygiene and safe food handling practices with a score of 100%.

## Association among food safety knowledge, attitudes and practices of food handlers

Significant correlations (table 6) were found between knowledge and attitudes (p=0001), knowledge and practices (p<0001), and attitudes and practices (p=0.02). However, the correlations between knowledge versus attitude and attitude versus practice were poor (Spearman's r<0.3), and the association between knowledge versus practice was moderate (0.3–0.7).

## DISCUSSION

This study showed that food handlers in CFAPs had relatively good knowledge, attitudes and practices. This is despite most food handlers (87.7%) lacking training in food safety, with only 12.3% having attended at least one official training course on food safety and general hygiene within 5 years of the study. Staff retention appeared to be somewhat high, given that 72.2% of food handlers had more than 10 years of service experience, 60.7% of which were in the food service industry. All food handlers (100%) understood the importance of washing

**Table 3** Food handlers' knowledge of safe food handling in charitable food assistance programmes (n=252)

| Statement/question | True No (%) | False No (%) |
|---|---|---|
| 1. It is essential to wash hands before handling food | 252 (100) | 0 (0) |
| 2. Wiping cloths can spread microorganisms | 208 (82.5) | 44 (17.5) |
| 3. The same cutting board can be used for raw and cooked foods, provided it looks clean | 39 (15.5) | 213 (84.5) |
| 4. Raw food needs to be stored separately from cooked food | 216 (85.7) | 36 (14.3) |
| 5. Cooked foods do not need to be thoroughly reheated | 186 (73.8) | 66 (26.2) |
| 6. Proper cooking includes meat cooked to 40°C | 233 (92.5) | 19 (7.5) |
| 7. Cooked meat can be left at room temperature overnight to cool before refrigerating | 62 (24.6) | 190 (75.4) |
| 8. Cooked food should be kept very hot before serving | 148 (58.7) | 104 (41.3) |
| 9. Refrigerating food only slows bacterial growth | 154 (61.1) | 98 (38.9) |
| 10. Safe water can be identified by the way it looks | 229 (90.9) | 23 (9.1) |
| 11. Wash fruits and vegetables | 251 (99.6) | 1 (0.4) |

**Table 4** Attitudes of food handlers towards safe food handling in charitable food assistance programmes (n=252)

| Statement/question | Agree No (%) | Unsure No (%) | Disagree No (%) |
|---|---|---|---|
| 1. Frequent handwashing during food preparation is worth the extra time | 252 (100) | 0 (0) | 0 (0) |
| 2. Keeping kitchen surfaces clean reduces the risk of illness | 252 (100) | 0 (0) | 0 (0) |
| 3. Keeping raw and cooked food separate helps to prevent illness | 225 (89.3) | 27 (10.7) | 0 (0) |
| 4. Using different knives and cutting boards for raw and cooked foods is worth the extra effort | 225 (89.3) | 14 (5.5) | 13 (5.2) |
| 5. Meat thermometers help ensure food is cooked thoroughly | 196 (77.8) | 29 (11.5) | 27 (10.7) |
| 6. Soups and stews should always be boiled to ensure safety | 244 (96.8) | 0 (0) | 8 (2.3) |
| 7. Thawing food in a cool place is safer | 244 (96.8) | 8 (2.3) | 0 (0) |
| 8. I think it is unsafe to leave cooked food out of the refrigerator for more than 2 hours | 189 (75.0) | 28 (11.1) | 35 (13.9) |
| 9. Inspecting food for freshness and wholesomeness is valuable | 252 (100) | 0 (0) | 0 (0) |
| 10. I think it is important to throw away foods that have reached their expiry date | 252 (100) | 0 (0) | 0 (0) |

their hands before handling food. Although 85% of food handlers knew the risks of food cross-contamination, 17% were unaware that wiping cloths can spread bacteria, mainly when used filthy. Almost all (92.5%) food handlers participating in this study did not realise the importance of thorough cooking and temperature control, as evidenced by 53% of food handlers who admitted to never properly reheating cooked meals. Although almost all food handlers (99.6%) reported that they wash fruits and vegetables, the majority (90.7%) thought one could tell water safety by simply looking at it.

According to this study, food handlers in the CFAPs recounted good knowledge, positive attitudes and good self-reported practices related to safe food handling. The findings are consistent with other studies conducted in the conventional food supply chain in LMIC settings.[26–30] In their recent study, Fariba et al[26] found that food handlers were good at separating raw and cooked materials and using clean water and cloths. Their findings are consistent with ours, revealing that most (85%) food handlers in CFAPs knew the risks of food cross-contamination. This is likely due to extensive food handling experience accrued over a long time, as most food handlers had over 10 years of experience, especially in the food service industry. When interviewed, some food handlers asserted that they had worked for big corporate restaurants before

**Table 5** Self-reported practices of food handlers in charitable food assistance programmes (n=252)

| Statement/question | Always No (%) | Mostly No (%) | Sometimes No (%) | Rarely No (%) | Never No (%) |
|---|---|---|---|---|---|
| 1. I wash my hands before and during food preparation | 252 (100) | 0 (0) | 0 (0) | 0 (0) | 0 (0) |
| 2. I clean surfaces and equipment used for food preparation before reusing them on other food | 252 (100) | 0 (0) | 0 (0) | 0 (0) | 0 (0) |
| 3. I use separate utensils and cutting boards when preparing raw and cooked food | 166 (65.9) | 59 (23.1) | 0 (0) | 0 (0) | 27 (11) |
| 4. I separate raw and cooked food during storage | 214 (84.9) | 9 (3.6) | 0 (0) | 0 (0) | 29 (12) |
| 5. I check that meats are cooked thoroughly by ensuring that the juices are clear or by using a thermometer | 217 (86.0) | 0 (0) | 0 (0) | 0 (0) | 35 (14) |
| 6. I reheat cooked food until it is piping hot throughout | 110 (43.7) | 6 (2.4) | 0 (0) | 0 (0) | 134 (53) |
| 7. I thaw frozen food in the refrigerator or other cool places | 140 (55.6) | 7 (28) | 23 (9.1) | 0 (0) | 82 (33) |
| 8. After I have cooked a meal, I store any leftovers in a cool place within 2 hours | 172 (68.3) | 2 (0.8) | 57 (22.6) | 0 (0) | 21 (8) |
| 9. I check and throw away food beyond its expiry date | 252 (100) | 0 (0) | 0 (0) | 0 (0) | 0 (0) |
| 10. I wash fruits and vegetables with safe water before eating them | 252 (100) | 0 (0) | 0 (0) | 0 (0) | 0 (0) |

**Table 6** Association among food safety knowledge, attitudes and practices

| Level | Spearman's r | Significance |
| --- | --- | --- |
| Knowledge versus attitude | 0.25 | 0.001 |
| Knowledge versus practice | 0.43 | <0001 |
| Attitude versus practice | 0.15 | 0.02 |

joining the CFAPs as food service staff. Their knowledge and experience can be enhanced with current and formal routine food safety training. Local authorities can support CFAPs by rendering services such as regular inspections, food sampling, certification/permitting, seizure and disposal of expired foodstuffs, and health and hygiene education and training, as seen in HICs.[31–33]

The current study's findings support prior research,[25–28] revealing that most food handlers in CFAPs are reckless with donated food. This is because while volunteering, no training in proper food handling is provided to food handlers. However, it was discovered that most food handlers (72.2%) had more than 10 years of experience and that 60.7% of the workers had spent most of their working lives in the food service industry. While this may appear to be good because workers have gained a lot of experience over time, several studies have warned against complacency as the job becomes dull and workers lose interest in their profession. Most food handlers become casual with safe food handling at this point, putting the food in danger of contamination.[34–36] Significant correlations were found between knowledge and attitudes (p=0001), knowledge and practices (p<0001), and attitudes and practices (p=0.02). However, the correlations between knowledge versus attitude and attitude versus practice were poor (Spearman's r<0.3), and the association between knowledge versus practice was moderate (0.3–0.7). This means that it is impossible to say whether respondents' food safety knowledge will influence their attitude, though it may influence their practices around food safety. The findings also show that food handlers' attitudes toward food safety do not accurately reflect their actual food safety practices.

On the other hand, some CFAPs' food handlers claimed a lack of basic infrastructure and resources, such as limited workstations, poor food preparation/processing equipment and inadequate storage facilities, to name a few. The walkthrough inspections and observations by the researcher also corroborated the reports by some CFAP food handlers. These challenges make it difficult for CFAP's food handlers to put their knowledge and competence acquired over the years to optimal use. Nevertheless, CFAP food handlers' understanding of proper reheating of cooked food, distinguishing safe water by appearance, food holding temperature and the danger zone remained weak. These findings were consistent with the conclusions made by Sharif et al,[37] asserting

that despite good knowledge levels among food handlers, they had insufficient knowledge on thawing, thorough cooking, reheating and safe temperature. Other studies have shown that this could pose potential risks or cause food poisoning in consumers.[24 38–40]

The study's key weakness is the lack of similar studies published in South Africa. The sample size of the study was not too large. However, the diverse CFAPs from which the sample was drawn offer unique strength to the study. However, our study was prone to social desirability bias, given that most data emanated from self-reporting. However, the infusion of a walkthrough as part of the data collection technique to complement the survey questionnaire, alongside the comprehensive nature of the consent process, served as essential mitigation. Nonetheless, the current findings can be used to support the promotion of food handler training, particularly in CFAPs. Based on our results, we suggest that future research investigate the perceived barriers and facilitators to food safety and general hygiene in CFAPs.

## CONCLUSION

Despite the relatively positive knowledge, attitude and practice levels of the food handlers, safe food handling and hygiene practices, such as thorough cooking and temperature control, properly reheating cooked meals and taking precautions to prevent cross-contamination, require some emphasis. To reduce the risk of food-borne illness, the CFAPs' food service workers should get regular health and hygiene instruction as well as food safety training and relevant manuals.

**Acknowledgements** IREX funding through the UASP alumni small grants programme is acknowledged for providing a platform for developing this publication. The authors would like to recognise the College of Health Sciences at the University of KwaZulu-Natal and eThekwini Municipality for all the valuable support.

**Contributors** SEM is responsible for the overall content as the guarantor. He conceptualised the study and wrote the manuscript. KH and MM contributed to the writing by critically reviewing the manuscript and making revisions. SEM wrote the final draft manuscript, and all the authors approved it.

**Funding** The authors have not declared a specific grant for this research from any funding agency in the public, commercial or not-for-profit sectors.

**Competing interests** None declared.

**Patient and public involvement** Patients and/or the public were not involved in the design, or conduct, or reporting, or dissemination plans of this research.

**Patient consent for publication** Not required.

**Ethics approval** This study involves human participants and ethical clearance (BREC/00000821/2019) was applied for and obtained from the University of KwaZulu-Natal Biomedical Research Ethics Committee (BREC). The eThekwini District approved for the study to be conducted, and all participants consented to participate in the study.

**Provenance and peer review** Not commissioned; externally peer reviewed.

**Data availability statement** Data are available upon reasonable request. Extra data can be accessed via the Dryad data repository at http://datadryad.org/ with the doi: 10.5061/dryad.g1jwstqvd.

**ORCID iD**
Sizwe Earl Makhunga http://orcid.org/0000-0001-8661-3075

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
