## [Reviewer comments · BMJ Open]

ARTICLE DETAILS

TITLE (PROVISIONAL)	Cross-sectional study on food handlers' knowledge, attitudes, and self-reported practices regarding safe food handling in charitable food assistance programs in the eThekweni District, South Africa
AUTHORS	Makhunga, Sizwe; Macherera, Margaret; Hlongwana, Khumbulani

VERSION 1 – REVIEW

REVIEWER	Meseret, Maru Debre Markos University College of Health Science, Department of Health Informatics
REVIEW RETURNED	13-Aug-2022

GENERAL COMMENTS	Comments to Food handlers' knowledge, attitudes, and self-reported practices of safe food handling in charitable food assistance programs in the eThekweni District, South Africa: A. Abstract 1. Line 28-29 "A criterion-based sampling technique and the exponential non-discriminative snowball sampling methods were used to identify the CFAPs". How can a non-scientific audience understand this design? Even you didn't operationalize it. 2. Line 29-30 "The principal investigator administered the validated English or isiZulu structured Questionnaire adapted from the World Health Organization". How did you control researcher bias? 3. Line 33-34 "The secondary outcomes were walkthrough surveys of infrastructure and observations of practices". B. Strengths and limitations 1. How can "the study's key strength is the lack of similar studies published in a South African setting" and "the current findings can be used to support the promotion of food handler training, particularly in CFAPs" line 49 and 55 respectively be the strengths of the study? C. Introduction 1. Line 102-105 "Despite several CFAPs operating in the eThekweni district for decades, there is a lack of data about the status of their safety and hygiene practices. Studies, however, have shown that the majority of CFAP
---

	food handlers are unskilled and operate on a volunteer basis" tells us that the skills of CFAP food handlers is already known. So, what is the significance of repeating it again? Otherwise, tell us the difference between skill and knowledge, attitudes, and practices toward safe food handling. D. Methods  1. Study Design line 138 is not clear 2. Why snowballing was used because the case is not rare 3. Line 149-150 is telling us that the data was collected by the principal investigator as described under abstract. How did you control investigator bias? 4. In the data collection section you described that you did a pilot study but you failed to tell us the outcome of the pilot study. Moreover, you told us that "each participant signed an informed consent form before participating in the study" but failed to annex the form. 5. It would be very nice if you used a model other than binary LR E. Result  1. You used a descriptive analysis. For me, it would be very nice if you ran some kind of model. Otherwise, your narration is very good.
--	--

REVIEWER	Amenu , Kebede Addis Ababa University
REVIEW RETURNED	13-Oct-2022

GENERAL COMMENTS	This study fairly reports about food handlers' knowledge, attitudes, and self-reported practices in charitable food assistance programs which is a unique context making it useful for readers and those working in food donation areas. The study can be useful for further actions given that those dependent on food donation can be potentially vulnerable groups. However, the study has some issues which demand revisions. Define CFAPs and use them consistently Throughout the paper: Use full stops instead of commas to separate decimals (English style). Probably, the computer machine was set to French mode.... Or corrected manually throughout. Knowledge, attitude and practices scores (the main objective of the study) were not presented in the abstract Lines 182-184: The statement is clear. I am not sure how Chi-square is used to calculate frequency. Chi-square is the test of independency The descriptions presented from lines 189-194 were not presented in the results section. For example, how many of the respondents had poor knowledge and similar question about attitudes and practices? Lines 196-203: These two parts are repeated also at the end of this manuscript.
--

	Limes 270-74: looks like the discussion. Rephrase or take it to discussion For your conclusion: try to get an idea from what has been reported in the paper rather than extrapolating.
--	--

VERSION 1 – AUTHOR RESPONSE

REVIEWER ONE QUERIES/ COMMENTS AND RESPONSES			
A. ABSTRACT			
1	Line 28-29 "A criterion-based sampling technique and the exponential non-discriminative snowball sampling methods were used to identify the CFAPs". How can a non-scientific audience understand this design? Even you didn't operationalize it.	We welcome and note the suggestion with thanks. The purpose was to keep the word count in this section to a minimum, as per the journal requirement. However, we have generally addressed this concern in the revised manuscript.	Page 1, Line 28 and Page 4, Line 149
2	Line 29-30 "The principal investigator administered the validated English or isiZulu structured Questionnaire adapted from the World Health Organization". How did you control researcher bias?	Thank you for your comment. We have revised this section.	Page 1, Lines 32 - 35.
3	Line 33-34 "The secondary outcomes were walkthrough surveys of infrastructure and observations of practices".	Thank you for your comment. We have revised this section.	Page 1, Lines 38 – 40.
B. STRENGTHS AND LIMITATIONS			
1	How can "the study's key strength is the lack of similar studies published in a South African setting" and "the current findings can be used to support the promotion of food handler training, particularly in CFAPs" line 49 and 55 respectively be the strengths of the study?	Thank you for your comment. We have revised this section.	Page 2, Lines 60 – 66.
C. INTRODUCTION			
1	Line 102-105 "Despite several CFAPs operating in the eThekweni district for decades, there is a lack of data about the status of their safety and hygiene practices. Studies, however, have shown that the majority of CFAP food handlers are unskilled and operate on a volunteer	Thank you for your comment. We have revised this section.	Page 3, Lines 117 – 119.

	basis" tells us that the skills of CFAP food handlers is already known. So, what is the significance of repeating it again? Otherwise, tell us the difference between skill and knowledge, attitudes, and practices toward safe food handling.		
D. METHODS			
1	Study Design line 138 is not clear	Thank you for your comment. We have revised this section.	Page 4, Line 149
2	Why snowballing was used because the case is not rare	The charitable food assistance practice is a rarely documented phenomenon in South Africa. The control of their activities by the government is fragmented and not simply regulated. Finding a database of food donors proved difficult because of this. Snowballing was used to build a database as we go. However, we thank you for your comment and have revised the section.	Page 1, Line 28 and Page 4, Line 149.
3	Line 149-150 is telling us that the data was collected by the principal investigator as described under abstract. How did you control investigator bias?	Thank you for your comment. We have revised this section.	Page 5, Line 180.
4	In the data collection section you described that you did a pilot study but you failed to tell us the outcome of the pilot study. Moreover, you told us that "each participant signed an informed consent form before participating in the study" but failed to annex the form.	We thank you for your comment. However, we did clarify that the information obtained from the pilot study was not included in the final analysis. They were merely for testing validity and reliability of the research tool (questionnaire). We have incorporated the results of the Cronbach's alpha test, which are acceptable at 0.76. We also have annexed the form, thank you.	Page 5, Lines 186-189.
5	It would be very nice if you used a model other than binary LR	We appreciate the comment. We shall consider your point for future endeavours.	
E. RESULTS			
1	You used a descriptive analysis. For me, it would be very nice if you ran some kind of model. Otherwise, your narration is very good.	We appreciate the positive remarks. We will certainly consider your comments/input for future works.	

REVIEWER TWO QUERIES/ COMMENTS AND RESPONSES

1	GENERAL COMMENT: This study fairly reports about food handlers' knowledge, attitudes, and self-reported practices in charitable food assistance programs which is a unique context making it useful for readers and those working in food donation areas. The study can be useful for further actions given that those dependent on food donation can be potentially vulnerable groups. However, the study has some issues which demand revisions.		
2	Define CFAPs and use them consistently	We have generally addressed this comment in the revised manuscript.	
3	Throughout the paper: Use full stops instead of commas to separate decimals (English style). Probably, the computer machine was set to French mode.... Or corrected manually throughout.	Thank you for your comment. Although there isn't a universally accepted decimal separator standard in Africa, some nations including Zimbabwe, South Africa, and Cameroon use a decimal comma style.	
4	Knowledge, attitude, and practices scores (the main objective of the study) were not presented in the abstract	Thank you for the comment. We have addressed this comment in the revised manuscript.	Page 1, Line 41
5	Lines 182-184: The statement is clear. I am not sure how Chi-square is used to calculate frequency. Chi-square is the test of independency	Thank you for the comment. We have addressed this comment in the revised manuscript.	Page 6, Line 207.
6	The descriptions presented from lines 189-194 were not presented in the results section. For example, how many of the respondents had poor knowledge and similar question about attitudes and practices?	Thank you for the comment. We have generally addressed this comment in the revised manuscript.	Page 10, Line 274; Page 10, Line 287 and Page 11, Line 300.
7	Lines 196-203: These two parts are repeated also at the end of this manuscript.	Thank you. We have addressed this comment in the revised manuscript by removing the repetition.	
8	Lines 270-74: looks like the discussion. Rephrase or take it to discussion	We appreciate the observation and the advice. We have addressed this comment in the revised manuscript.	Page 13, Line 351.
9	For your conclusion: try to get an idea from what has been reported in the paper rather than extrapolating.	Thank you. We have addressed this comment in the revised manuscript.	Page 14, Line 383.

--	--	--	--

VERSION 2 – REVIEW

REVIEWER	Amenu , Kebede Addis Ababa University
REVIEW RETURNED	04-Feb-2023

GENERAL COMMENTS	Some language editions are needed which can be addressed during the proofreading, I hope.
---